# A Hybrid Model Based on Principal Component Analysis, Wavelet Transform, and Extreme Learning Machine Optimized by Bat Algorithm for Daily Solar Radiation Forecasting

**Xing Zhang \* and Zhuoqun Wei**

Department of Business Administration, North China Electric Power University, Baoding 071000, China
\* Correspondence: 51851719@ncepu.edu.cn

**Abstract:** Precise solar radiation forecasting is of great importance for solar energy utilization and its integration into the grid, but because of the daily solar radiation's intrinsic non-stationary and nonlinearity, which is influenced by a lot of elements, single predicting models may have difficulty obtaining results with high accuracy. Therefore, this paper innovatively puts forward an original hybrid model that predicts solar radiation through extreme learning machine (ELM) optimized by the bat algorithm (BA) based on wavelet transform (WT) and principal component analysis (PCA). First, choose the meteorological variables on the basis of Pearson coefficient test, and WT will decompose historical solar radiation into two time series, which are de-noised signal and noise signal. In the approximate series, the lag phase of historical radiation is obtained by partial autocorrelation function (PACF). After that, use PCA to reduce the dimensions of the influencing factors, including meteorological variables and historical radiation. Finally, ELM is established to predict daily solar radiation, whose input weight and deviation thresholds gained optimization by BA, thus it is called BA-ELM henceforth. In view of the four distinct solar radiation series obtained by NASA, the empirical simulation explained the hybrid model's validity and effectiveness compared to other primary methods.

**Keywords:** solar radiation forecasting; ELM; BA; WT; PACF; PCA; Pearson coefficient test

## 1. Introduction

Daily total solar radiation is considered as the most important parameter in meteorology, solar conversion, and renewable energy applications, particularly for the sizing of stand-alone photovoltaic (PV) systems. The knowledge of the amount of solar radiation falling on the surface of the earth is of prime importance to engineers and scientists involved in the design of solar energy systems. In particular, many design methods for thermal and photovoltaic systems require information about the daily radiation on a horizontal surface in order to predict the energy production of the system, and its prediction precision is extremely instructive for the stable operation of the power grid as well as the formulation of a scheduling plan [1].

Many researchers have produced daily solar radiation (DSR) predictions. DSR prediction methods are usually divided into three categories: conventional physical models, mathematical statistical models, and machine learning. Conventional physical models predict solar radiation values through a series of physical analysis, data fitting, complex mathematical model construction, and calculations in the absence of meteorological data on total solar radiation intensity. A lot of authors have succeeded in various clear day models as well, including the simple ones of the half-sine [2,3] and the Collares-Pereira

and Rabl model [4]. Physical models do not reflect the strong randomness of the solar radiation sequence. Once the meteorological environment changes, the calculation accuracy is greatly reduced.

The traditional mathematical statistics models include regression analysis [5], time series analysis [6,7], gray theory [8], fuzzy theory [9,10], and Kalman filter [11]. Trapero et al. (2015) [5] applied dynamic harmonic regression model (DHR) to forecast short-term direct solar radiation and scattered solar radiation in Spain for the first time in 2015. Huang et al. (2013) [6] used the autoregressive model to predict the 2013 meteorological factors—when the solar radiation is within the dynamic adjustment system framework, the accuracy is 30% higher than the general neural network or stochastic model. Through the integration of Fourier transform and neural network, Fidan et al. (2014) [8] predicted hourly solar radiation in Izmir, Turkey. Olcan, Mahmudov (2016) [10] proposed a modified fuzzy time series (FTS) using eight different radiation mixing models. The results show that, compared with other fuzzy models and traditional time series methods, the proposed hybrid model-8 exhibits better performance. Akarslan et al. (2014) [11] first utilized the multi-dimensional linear predictive filtering model to predict solar radiation, and the two-dimensional linear predictive filtering model as well as the traditional statistical forecasting method have passed through empirical analysis.

With the development of big data mining, machine learning technology has attracted widespread attention. For instance, artificial neural network (ANN) [12–17] and support vector machine (SVM) [18–20] have been widely applied in solar radiation prediction. Amrouche and Le Pivert (2014) [12] took advantage of spatial modeling and artificial neural networks (ANNs) to predict daily total solar radiation in four locations in the United States, and the empirical results indicate that the proposed model satisfies the expected accuracy. Benmouiza and Cheknane (2013) [13] used K-means to classify the input data, then used nonlinear autoregressive neural networks to model various categories, and eventually predicted the solar radiation of test data through the accordant model. Adel, Massi (2010) [15] used artificial neural network (ANN) for solar irradiance prediction. A comparison between the forecasted one and the energy produced by the Grid Connected Photovoltaic Plants (GCPV) installed on the rooftop of the municipality of Trieste showed the advantages of the model. From the above, the conclusion can be drawn that, when the data set is not enough, ANN cannot perform well. Ekici, B.B. [18] developed an intelligent model based on least squares support vector machine (LSSVM) to forecast solar radiation for the next day. The number of days from 1 January, the daily average temperature, the daily maximum temperature, the sunshine time, and the sun day before the parameter were used as input to predict the daily sun sunshine. The results indicated that LSSVM is a superb approach to evaluate the amount of solar radiation of a specific location with an accuracy of 99.294%. Sun Shaolong et al. [19] put forward a decomposition cluster set (DCE) learning method for solar radiation prediction. In the proposed DCE learning method, (1) Ensemble Empirical Mode Decomposition (EEMD) is used to decompose the original solar radiation data into several intrinsic mode functions (IMF) and residual components, (2) least squares support vector regression (LSSVR) is utilized to predict IMF and residual component, and (3) the Kmeans method is used to cluster the prediction results of all components. The empirical analysis of the solar radiation data introduced in Beijing shows that, compared with other benchmark models, the Normalized Root Mean Square Error (NRMSE) and Mean Absolute Percentage Error (MAPE) generated by the DCE learning method are smaller, and the accuracy rates are 2.96% and 2.83%, respectively. In the forecast one day ahead, Meenal and Selvakumar [20] assessed the accuracy of support vector machine (SVM), artificial neural network (ANN), and empirical solar radiation models with different combinations of input parameters. The parameters include month, latitude, longitude, bright sunshine hours, day length, relative humidity, and maximum and minimum temperature. Based on statistical measures, the daily solar radiation forecasting models of different cities in India were evaluated. The results indicated that, compared with ANN and empirical models, the SVM model with the most influencing input parameters is superior. However, diverse types of kernel functions and kernel parameters greatly affect the accuracy of fitting and generalization.

The Extreme Learning Machine (ELM) originally put forward by Huang in 2004 [21] has faster convergence speed and less human interference than traditional neural networks and can also prevent possible occurrences in gradient-based learning, such as stopping criteria, learning rate, and learning periods. In view of this, extreme learning machines are widely used in different forecasting areas, load forecasting [22,23], wind speed forecasting [24,25], electricity price forecasting [26], carbon emission forecasting [27], and so on. However, the input weight matrix and hidden layer bias of the randomly assigned ELM is likely to influence the generalization ability of the ELM. Therefore, an optimization algorithm is highly desirable to obtain the optimal weight of the input layer and the bias of the hidden layer.

The Bat Algorithm (BA) [28] is considered to be a new meta-heuristic method that dynamically controls the mutual conversion between local search as well as global search and achieves better convergence. For the superb performance of local search and global search compared with existing algorithms such as genetic algorithm (GA) [29] and particle swarm optimization (PSO) [30], researchers and scholars have made wide use of BA in various optimization problems [31–33]. Qi Liu et al. [31] proposed a novel Hybrid Bat Algorithm for complex continuous optimization problems. Deepak Gupta et al. [32] proposed an Optimized Binary Bat algorithm for classification of different types of leukocytes. Lili A et al. [33] proposed bat algorithm to minimize total generator cost from the thermal power plant, and their experimental results showed that bat algorithm is able to save approximately 1.23% compare to the actual cost.

Therefore, this paper optimizes the input weight and hidden bias of the extreme learning machine through BA to realize the advantages of maximizing the global and local search capabilities of BA and the goal of ELM fast learning speed, which overcomes the inherent instability of ELM.

Considering the solar radiation's inherent complexity, which is influenced by many parameters, it is expected to complete data processing ahead [34,35]. Wavelet transform (WT) is considered to be the most commonly used data preprocessing method for decomposing time series and eliminating stochastic volatility. Tan et al. [36] succeeded in using wavelet decomposition to decompose the electricity price sequence into an approximate sequence and detail sequences, and each sub-series can be forecasted separately by an appropriate time series model. The results show that WT can capture the complex features of non-stationary, nonlinear, and high volatility.

Based on the above studies, it can be found that the appropriate selection of influencing factors has a significant impact on the prediction of solar radiation. Nevertheless, most studies only emphasize the effects of these factors on solar radiation, ignoring the interrelationships between them. In fact, the information contained in the data overlaps, so the computational efficiency is greatly reduced due to the complexity of the network. Principal component analysis (PCA) simplifies the network structure and significantly improves operational efficiency and prediction accuracy by minimizing the dimensionality of pre-selected influencing factors for information retention. Sun, Wet al. [37] used PCA to draw original features and the dimensions of the LSSVM input selection were reduced to predict daily PM2.5 concentrations. Experimental studies show that this method is superior to the single LSSVM model. Therefore, PCA is utilized in this paper with the intention of reducing the dimensionality of data and improving prediction accuracy.

At present, most of the empirical studies on solar radiation prediction use data from a certain region or a certain country. Few use different longitudes and different dimensions to predict simultaneously. In order to verify the solar radiation prediction model proposed in this paper. The validity and application of four solar radiation time series in Beijing (40 degrees north latitude 116), New York (north latitude 40 degrees −73), Melbourne (latitude −37, longitude 145), and São Paulo (latitude −23, longitude −46) are studied in this paper.

In summary, after the WT decomposition, the solar radiation series is split into an approximate series and a detailed series. Then, the detailed time series is discarded, and the approximate time series and meteorological indicators are processed by partial autocorrelation analysis (PCA) to further determine the input variables of the prediction model. Finally, a BA-optimized Extreme Learning

Machine (ELM) is applied to obtain predicted daily solar radiation. In order to verify the validity and superiority of the proposed model, four different sites were simulated in this paper. The main contributions of this article are as follows.

- The factors affecting solar radiation contain meteorological indicators and historical data on solar radiation in this paper;
- ELM is a new type of neural network that has been applied in solar radiation prediction, which avoids the shortcomings of slow learning, large training samples, and over-fitting in previous studies;
- The BA-optimized ELM application further improves the robustness and prediction accuracy of the model;
- Implementation of WT greatly reduces the difficulty of solar radiation prediction;
- This paper focuses on the correlation between influencing factors and uses PCA to reduce the dimensionality to improve computational efficiency and prediction accuracy;
- This may be the first paper to study solar radiation prediction methods that can be applied to different parts of the world at the same time.

The structure of this paper is as follows: Section 2 briefly introduces WT, PACF, PCA, BA, ELM, and BA-ELM, and the new hybrid prediction technique (PCA-WT-BA-ELM) is then discussed in detail. Section 3 provides empirical analysis, which includes data collection, input selection, parameter settings, prediction results, and error analysis for four cities. Section 4 shows the general conclusions based on the experimental results.

## 2. Methodology

### 2.1. Wavelet Transform

Wavelet decomposition and reconstruction are based on multi-resolution analysis. It was first proposed by Mallat in 2000 [38] and is one of the most useful tools in signal analysis. The observation data usually consists of two parts—the true value and the error (i.e., noise). The true value (i.e., the useful signal) in the observed data is different from the characteristic exhibited by the random noise in the time-frequency domain. The useful features of the useful signal in the time domain and the frequency domain are obvious, and generally appear as low frequency signals. The random noise has obvious global characteristics in the time domain and the frequency domain, and the high frequency signal appears in the frequency domain. According to the different characteristics of the two in the time-frequency domain, multi-resolution analysis by wavelet transform can be performed. The components of different frequencies are effectively separated to eliminate random noise. Finally, according to the inverse operation of wavelet transform, the denoising processing of the original observation data is realized by wavelet reconstruction [39].

The Wavelet Transform equation is defined as the integral of the signal multiplied by scaled, shifted versions of a basic wavelet function—a real-valued function whose Fourier Transform fulfill the admissibility criteria.

$$W_f(a,b) = \frac{1}{\sqrt{a}} \int f(t) \Psi\left(\frac{t-b}{a}\right) dt \tag{1}$$

where $a$ is the so-called scaling parameter and b is the time positioning or shifting parameter. Both $a$ and $b$ can be continuous or discrete variables. t represents time, $f(t)$ represents the original signal, and $\Psi\,(\cdot)$ represents the mother wavelet function. $W_f(a,b)$ is the result of the wavelet transform.

### 2.2. Bat Algorithm

Bat algorithm is inspired by micro-bats' echolocation behavior, through which bats are able to probe prey and evade obstacles. It has the advantages of parallelism, fast convergence, and less distribution and parameter adjustment [33].

In the d-dimensions of search space during the global search, the bat *i* has the position of $x_i^t$, and velocity $v_i^t$ at the time of t, whose position and velocity will be updated as Equations (2) and (3), respectively:

$$x_i^{t+1} = x_i^t + v_i^{t+1} \tag{2}$$

$$v_i^{t+1} = v_i^t + \left(x_i^t - x^\Lambda\right) \cdot F_i \tag{3}$$

where $x^\Lambda$ is the current global optimal solution. $F_i$ is the sonic wave frequency that can be seen in Equation (4).

$$F_i = F_{min} + (F_{max} - F_{min}) \cdot \beta \tag{4}$$

where $\beta$ is a random number within [0, 1] and $F_{max}$ and $F_{min}$ are the max and min sonic wave frequency of the bat *i*. In the flight, each initial bat is allocated one frequency in conformity with $[F_{min}, F_{max}]$ randomly.

If a solution is selected in the current global optimal solution in local search, each bat will bring about a new alternative solution in the way of random walk according to Equation (5).

$$x_n(i) = x_0 + \mu A^t \tag{5}$$

where $x_0$ is the solution which is randomly selected in current best disaggregation, $A^t$ represents the mean of current bat populations, and $\mu$ is the D-dimensional vector within [−1, 1].

Impulse volume $A(i)$ and impulse emission rate $R(i)$ controlled the balance of bats. When a bat aims at its prey, the volume $A(i)$ will declines while the emission rate R(i) ascends. The update of $A(i)$ and $R(i)$ are expressed as Equations (6) and (7), respectively.

$$A^{t+1}(i) = \gamma A^t(i) \tag{6}$$

$$R^{t+1} = R^0(i) \cdot \left(1 - e^{-\theta t}\right) \tag{7}$$

where $\gamma$ and $\theta$ are the attenuation coefficient of the volume and the enhancement factor of the search frequency, respectively. $\gamma$ is within [0, 1], and $\theta > 0$.

It has already been proven (Yang, 2012) [28] that bat algorithm is potentially more powerful than PSO, GA, and Harmony Search. Because BA uses a good combination of major advantages of these algorithms in some way, it has been confirmed by Yang that bat algorithm is potentially more powerful than PSO, GA, and Harmony Search. Thanks to its parallelism, quick convergence, distribution, and less parameter adjustment, BA has been utilized in various areas.

## 2.3. Extreme Learning Machine

ELM is a novel algorithm based on single hidden layer feed forward neural network (SFLN). Most traditional neural networks, as their nature of the gradient descent method, adjust the weight and bias through multiple iterations, making them slow for training and easy to plunge into the local optimum. Their performance is also subject to certain limitations because it is sensitive to the learning rate. On account of its high sensibility to the learning rate, their performance is restricted [21].

To improve the SLFN, ELM, as shown in Figure 1, randomly assign the weight of the input layer and the thresholds of the hidden layer. Without a necessary iteration, the speed of completing the network learning is accelerated. Once the number of the hidden nodes is set, ELM can make use of the Moore-Penrose (MP) [40] generalized inverse matrix to calculate the output weight, which transforms the training program into solving the least square problem. Moreover, ELM is more accurate on performance than other neural networks [41]. The calculation steps of the standard ELM can be explained as follows:

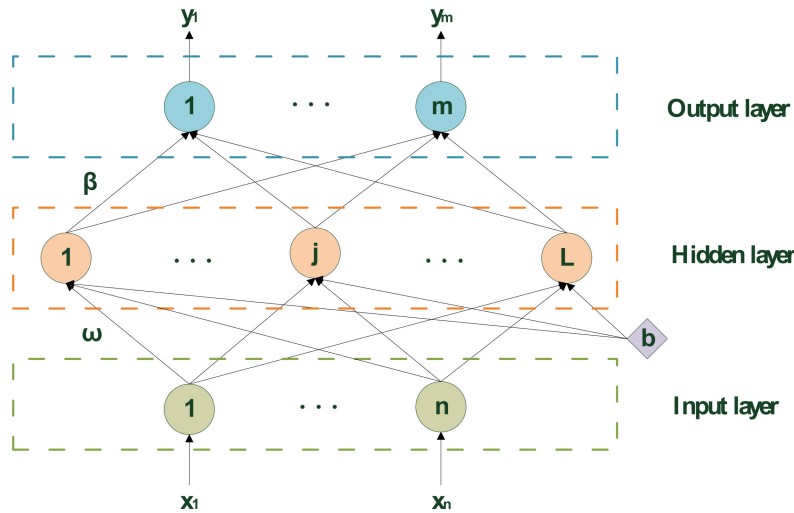

**Figure 1.** The framework of the extreme learning machine.

The ELM consists of an input layer including, an implicit layer, and an output layer, where the n is input layer neuron number, which corresponds to **n** input variables **x₁** ... **xₙ**. The hidden layer neuron number is L; the output layer neuron number is m, corresponding to m input variables y1 ... yₙ.

The connection weights between the input layer and the hidden layer is $\omega = \left[\omega_{ij}\right]_{n \times L}$, $i = 1L\ n$, $j = 1L\ L$, and the connection weight between hidden layer and output layer is. $\beta = \left[\beta_{jk}\right]_{L \times m}$, $j = 1L\ L, k = 1\ Lm$.

Make the training set input matrix with Q samples to be $X = \left[x_{ir}\right]_{n \times Q}$, $i = 1L\ n, r = 1\ LQ$ and the output matrix to be $Y = \left[y_{kr}\right]_{m \times Q}$, $k = 1L\ m, r = 1L\ Q$. The hidden layer neuron threshold is $b = \left[b_1, b_2L\ b_L\right]^T$, the hidden layer activation function is $g(x)$, and the expected output of the network is $T = \left[t_1, t_2L\ t_m\right]$. Therefore, ELM can be illustrated as

$$T' = \begin{bmatrix} t_1 \\ t_2 \\ M \\ t_m \end{bmatrix} = \begin{bmatrix} \sum_{j=1}^{L} \beta_{j1} g\left[\omega_j g x_i + b_j\right] \\ \sum_{j=1}^{L} \beta_{j2} g\left[\omega_j g x_i + b_j\right] \\ M \\ \sum_{j=1}^{L} \beta_{jm} g\left[\omega_j g x_i + b_j\right] \end{bmatrix} \qquad i = 1L\ n \qquad (8)$$

### 2.4. The Proposed Model

Despite the fact that ELM has expected performance in major cases, its weakness affects its accuracy. While learning, the possible non-optimal or unnecessary weight values as well as thresholds may reduce ELM's performance, leading to erratic results. Additionally, in some practical applications, ELM demands a lot of hidden layer nodes to receive expected results, which precisely adds complications and makes it easy to overfit.

In order to solve the above problems and obtain a stable network, an extreme learning machine based on the bat algorithm is proposed to guarantee that the input weight and the bias threshold are reasonably selected. The proposed model takes full advantage of BA's global search capability and ELM's rapid convergence rate and also overcomes the inherent problems of ELM. Consequently, BA-ELM performs better in generalization, function approximation, and has more stable simulation results.

Figure 2 shows the whole flowchart of daily solar radiation forecasting, which is divided into four parts.

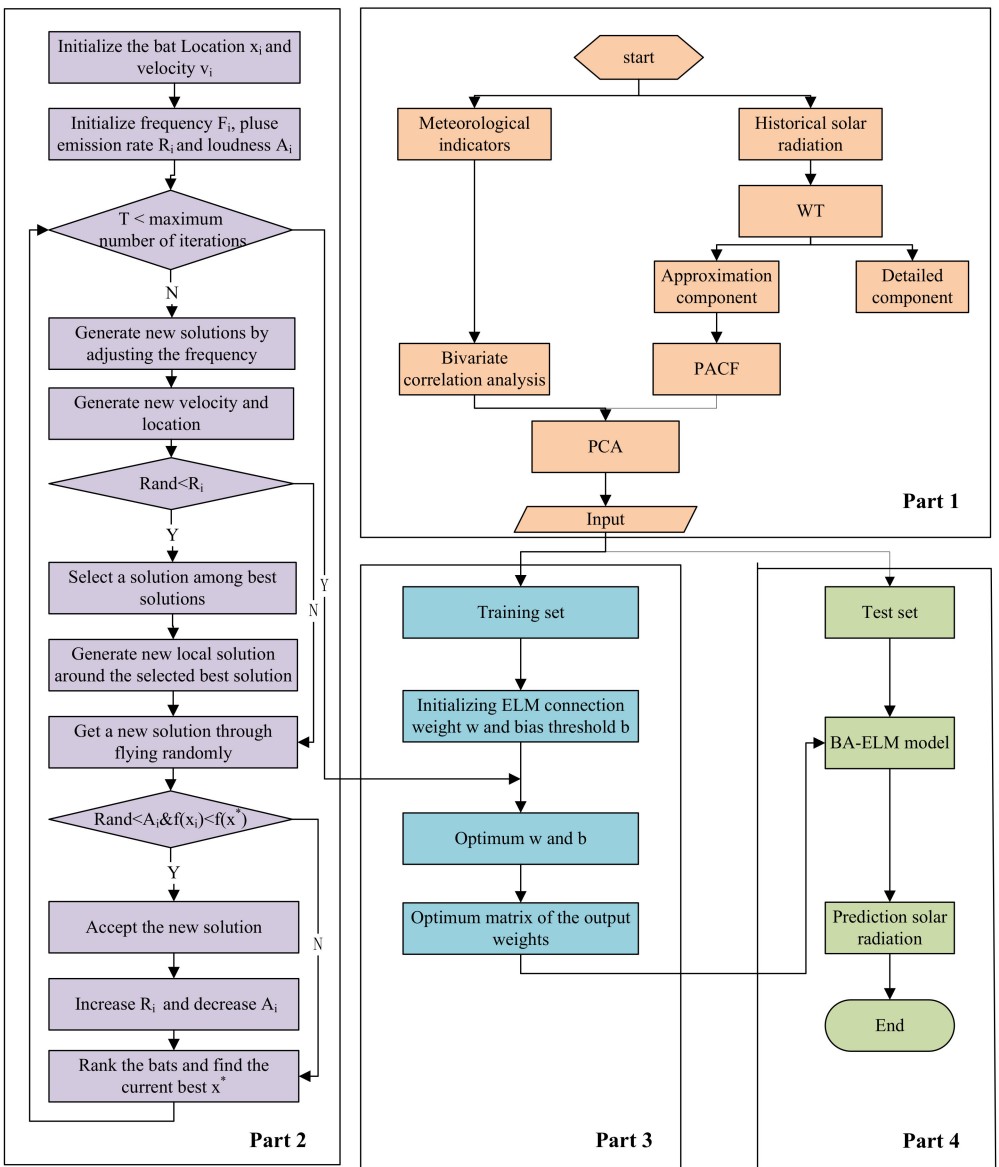

**Figure 2.** The flowchart of daily solar radiation forecasting model.

Part 1 is designed for input selection. First, the meteorological indicator is chosen in view of the Pearson coefficient test, and then the original historical solar radiation sequence is decomposed into two parts: an approximate series and a detailed series. The detailed series is abandoned, and PACF is applied to analyze the intrinsic relationships between the approximation series so as to determine the lag phases of historical radiation. PCA is used to decline the dimensionality of the influential factors, which contains meteorological indicators selected by Pearson coefficient test and historical radiation indicators selected by WT and PACF. The result is the inputs of BA-ELM.

Part 2 is the bat optimization algorithm (BA). It is obvious that BA is utilized to make the weight of the input layer and the bias of the hidden layer in ELM more optimal, for which the expected network can be achieved.

Part 3 is the training process of extreme learning machine (ELM). The training set data is derived from Part 1, and the parameters of ELM are optimized by Part 2, so that the ELM model is obtained with less training errors.

Part 4 is the testing process of ELM. The test set data is derived from Part 1, and the trained model is provided by Part 3 to obtain the test set prediction values.

## 3. Empirical Analysis

### 3.1. Data

Four typical cities are selected as research objects, which are Beijing (latitude 40, longitude 116), New York (latitude 40, longitude −73), Melbourne (latitude −37, longitude 145), and São Paulo (latitude −23, longitude −46). The four cities are at different latitudes and longitudes, which makes them more representative. North latitude and east longitude are positive, while the south latitude and west longitude is negative. The locations of the four cities are shown in Figure 3. The historical data of solar radiation and their meteorological influential indicators from 1 January 2014 to 31 December 2018 in four districts can be acquired from NASA [42].

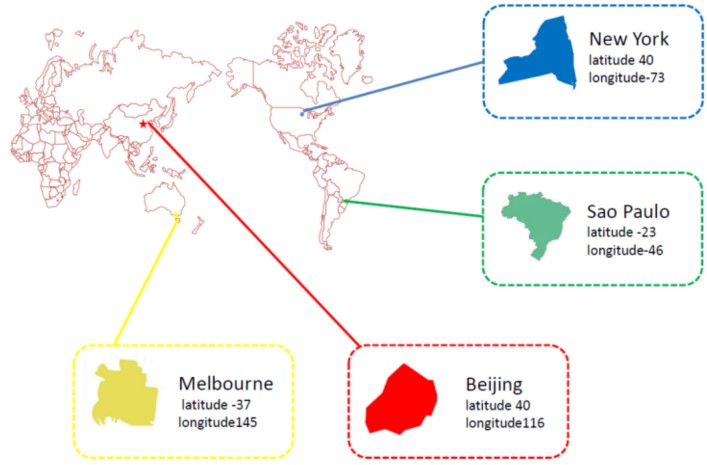

**Figure 3.** Locations of Beijing, New York, Melbourne, and São Paulo.

Figure 4 shows the daily solar radiation curves for the four regions, reflecting the highly uncertain, nonlinear, dynamic, and complex characteristics of solar radiation. The curve of Beijing and New York are similar, and the solar radiation peaks both appear in the summer. The curve of Melbourne and São Paulo are similar as well, and solar radiation peaks both appear in the winter because Beijing and New York are located in the north latitude, while Melbourne and São Paulo are located in the south latitude. The curve of New York is less volatile than that of Beijing, and the curve of São Paulo is less volatile than that of Melbourne. Therefore, there are four different types of solar radiation time series. This proposed model is applied to the prediction of these four solar radiation time series to prove its versatility and applicability.

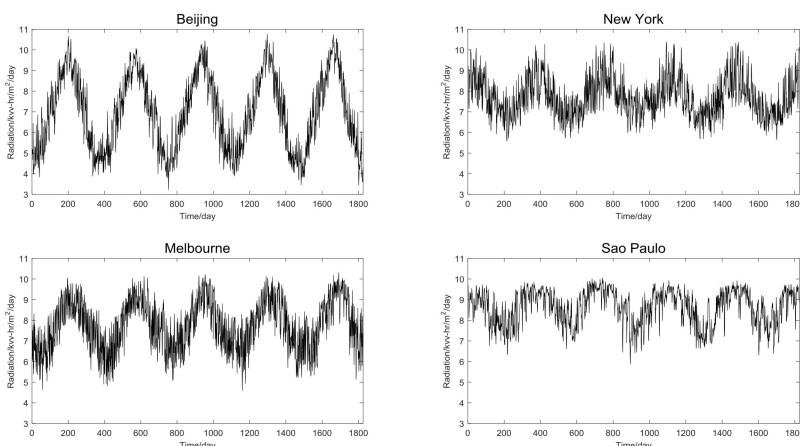

**Figure 4.** The original solar radiation data of Beijing, New York, Melbourne, and São Paulo.

The samples are divided into two subsets—the training set and the testing set. The training set includes the date from 1 January 2014 to 31 December 2017, which occupies about 80% of the entire date and is used for building prediction models. Another 20% of the data is testing set, which includes data from 1 January 2018 to 31 December 2018 for testing the accuracy of the established model. Taking the solar radiation date prediction of Beijing for instance, we confirm the model's function and superiority. The forecasting radiation date of New York, Melbourne, São Paulo is selected to demonstrate the model's validity.

### 3.2. Input Selection

3.2.1. Selection of Meteorological Indexes by Pearson Coefficient Test

Given that solar radiation is interfered with by many meteorological factors, it is critical to mine the relationships between solar radiation and the 21 pre-selected meteorological variables, for which an ideal prediction model can be established. Pearson coefficient test is chosen to conduct correlation analysis by SPSS 19.0. Table 1 presents the values of correlation coefficients.

**Table 1.** Pearson correlation coefficient of solar radiation and various meteorological indicators.

| Indicator | Abbreviation | Unit | Beijing | New York | Melbourne | São Paulo |
|---|---|---|---|---|---|---|
| Precipitation | PRECTOT | Mm/day | 0.118 ** | 0.089 ** | −0.020 | 0.101 ** |
| Specific Humidity at 2 M | SH2M | kg/kg | 0.911 ** | 0.883 ** | 0.834 ** | 0.895 ** |
| Relative Humidity at 2 M | RH2M | % | 0.446 ** | 0.562 ** | −0.446 ** | 0.502 ** |
| Surface Pressure | SP | kPa | −0.779 ** | −0.290 ** | −0.568 ** | −0.658 ** |
| Temperature Range at 2 M | T2M_RANGE | C | −0.162 ** | -0.449 ** | 0.237 ** | −0.572 ** |
| Earth Skin Temperature | EST | C | 0.922 ** | 0.768 ** | 0.800 ** | 0.691 ** |
| Dew/Frost Point at 2 M | T2M_DEW | C | 0.967 ** | 0.906 ** | 0.828 ** | 0.901 ** |
| Maximum Temperature at 2 M | T2M_MAX | C | 0.858 ** | 0.840 ** | 0.713 ** | 0.285 ** |
| Temperature at 2 M | T2M | C | 0.926 ** | 0.844 ** | 0.893 ** | 0.868 ** |
| Minimum Temperature at 2 M | T2M_MIN | C | 0.963 ** | 0.849 ** | 0.815 ** | 0.667 ** |
| Wind Speed Range at 50 M | WS50M_RANGE | m/s | −0.139 ** | −0.167 ** | 0.361 ** | 0.042 |
| Wind Speed Range at 10 M | WS10M_RANGE | m/s | −0.147 ** | −0.175 ** | 0.397 ** | 0.176 ** |
| Minimum Wind Speed at 50 M | WS50M_MIN | m/s | −0.408 ** | −0.119 ** | −0.076 ** | −0.085 ** |
| Minimum Wind Speed at 10 M | WS10M_MIN | m/s | −0.417 ** | −0.195 ** | −0.081 ** | −0.154 ** |
| Maximum Wind Speed at 50 M | WS50M_MAX | m/s | −0.422 ** | −0.234 ** | 0.224 ** | −0.035 |
| Maximum Wind Speed at 10 M | WS10M_MAX | m/s | −0.345 ** | −0.302 ** | 0.261 ** | 0.063 ** |
| Wind Direction at 50 M | WD50M | m/s | −0.471 ** | −0.368 ** | −0.041 | 0.194 ** |
| Wind Direction at 10 M | WD10M | m/s | −0.481 ** | −0.364 ** | −0.032 | 0.190 ** |
| Wind Speed at 50 M | WS50M | m/s | −0.457 ** | −0.197 ** | 0.109 ** | −0.053 * |
| Wind Speed at 10 M | WS10M | m/s | −0.426 ** | −0.278 ** | 0.154 ** | −0.073 ** |
| Insolation Clearness Index | ICI | | 0.048 * | 0.001 | 0.000 | −0.013 |

Notes: ** Significantly correlated at the 0.01 level (both sides). * Significantly correlated at the 0.05 level (both sides).

In order to show that there is a positive correlation between solar radiation and selected indicators, a meteorological index with a Pearson correlation coefficient greater than 0.8 is chosen as the factor. The selected meteorological indicators of Beijing, New York, Melbourne, and São Paulo are shown in Table 2. Consequently, it is important to take consideration of selected variables when predicting solar radiation.

**Table 2.** The selected meteorological indicators of Beijing, New York, Melbourne, and São Paulo.

| Region | Selected Indexes | Indexes Number |
|---|---|---|
| Beijing | SH2M, EST, T2M_DEW, T2M_MAX, T2M, T2M_MIN | 6 |
| New York | SH2M, EST, T2M_DEW, T2M_MAX, T2M, T2M_MIN | 6 |
| Melbourne | SH2M, EST, T2M_DEW, T2M_MAX, T2M, T2M_MIN | 6 |
| São paulo | SH2M, SP, EST,T2M_DEW, T2M, T2M_MIN | 6 |

### 3.2.2. Decomposition of Solar Radiation Series by WT

For the purpose of avoiding noise interference, WT is used to resolve the time series and remove random fluctuations. The parameters are set as follows: decomposition level = 1, Wavelet-basis function = 'db4', and WT is calculated in MATLAB R2017a. The data processing result of Beijing, New York, Melbourne, and São Paulo are presented in Figure 5.

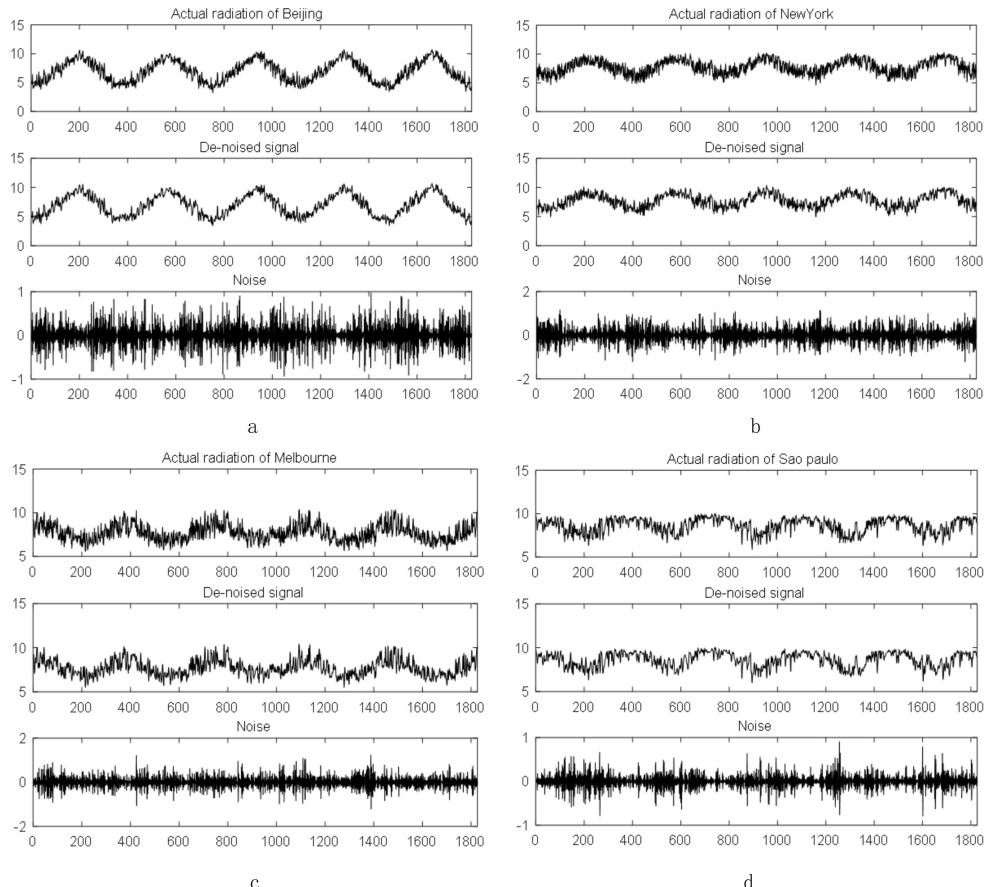

**Figure 5.** (**a**) The wavelet transform (WT) decomposed results of Beijing solar radiation series; (**b**) The WT decomposed results of New York solar radiation series; (**c**) The WT decomposed results of Melbourne solar radiation series; (**d**) The WT decomposed results of São Paulo solar radiation series.

As Figure 5 demonstrated, the solar radiation series are decomposed into de-noised signal (A1) and noise signal (D1). The main fluctuations in solar radiation are represented by de-noised signal, while noise signal contains spikes and stochastic volatility. When comparing with original solar radiation series, it is found that A1 provides a smooth form, while D1 represents a high frequency component. We take A1 as the solar radiation to model in order to improve efficiency.

### 3.2.3. Determination of the Lags by PACF

For the purpose of testing the correlation between historical radiation and radiation targets, this paper introduces some autocorrelation functions to choose the model's input variables. That is to say, the PACF calculated by SPSS 19.0 is employed to discover the Lags which are apparent after the cancellation of internal correlation. Figure 6 indicates the PACF results of de-noised signal of Beijing, New York, Melbourne, and São Paulo solar radiation series after WT.

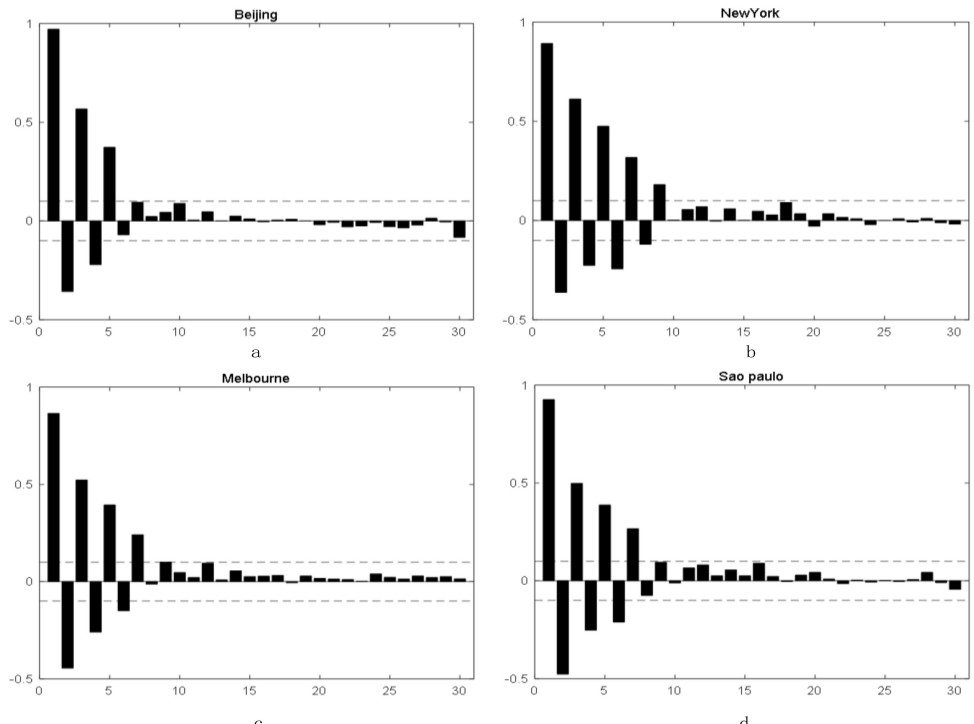

**Figure 6.** (**a**) Partial autocorrelation function (PACF) results of Beijing solar radiation series after WT; (**b**) PACF results of New York solar radiation series after WT; (**c**) PACF results of Melbourne solar radiation series after WT; (**d**) The PACF results of São Paulo solar radiation series after WT.

Set $x_i$ as the output variable and apply $x_{i-k}$ as one of the input variables if the PACF at lag k exceeds the 95% confidence interval. Table 3 presents the chosen variables of solar radiation in Beijing, New York, Melbourne, and São Paulo after WT.

**Table 3.** The lags determination of solar radiation by PACF after WT.

| City | Lag |
|---|---|
| Beijing | $(x_{t-1}, x_{t-2}, x_{t-3}, x_{t-4}, x_{t-5})$ |
| New York | $(x_{t-1}, x_{t-2}, x_{t-3}, x_{t-4}, x_{t-5}, x_{t-6}, x_{t-7}, x_{t-8}, x_{t-9})$ |
| Melbourne | $(x_{t-1}, x_{t-2}, x_{t-3}, x_{t-4}, x_{t-5}, x_{t-6}, x_{t-7})$ |
| São Paulo | $(x_{t-1}, x_{t-2}, x_{t-3}, x_{t-4}, x_{t-5}, x_{t-6}, x_{t-7})$ |

### 3.2.4. Reduction of Dimensionality by PCA

PCA is used to eliminate the multicollinearity that occurs in predictors, which is in view of the pre-selected variables in Sections 3.2.1 and 3.2.3. We receive the main information included in the data by using this approach. The PCA calculation was achieved on SPSS 19.0, and the results of Beijing, New York, Melbourne, and São Paulo are illustrated in Figure 6.

The red line in Figure 7 represents accumulated variance contribution rate, and the principal components whose accumulated variance contribution rate is more than 80% will be extracted. It can be discovered that the first major component of Beijing, the first major component of New York, the first three major components of Melbourne and the first two major components of São Paulo explain the elements accounting for more than 80%, so these major components are chosen as the BA-ELM input, as presented in Table 4, where $x_{t-i}$ represents the i-th lag of historical solar radiation data.

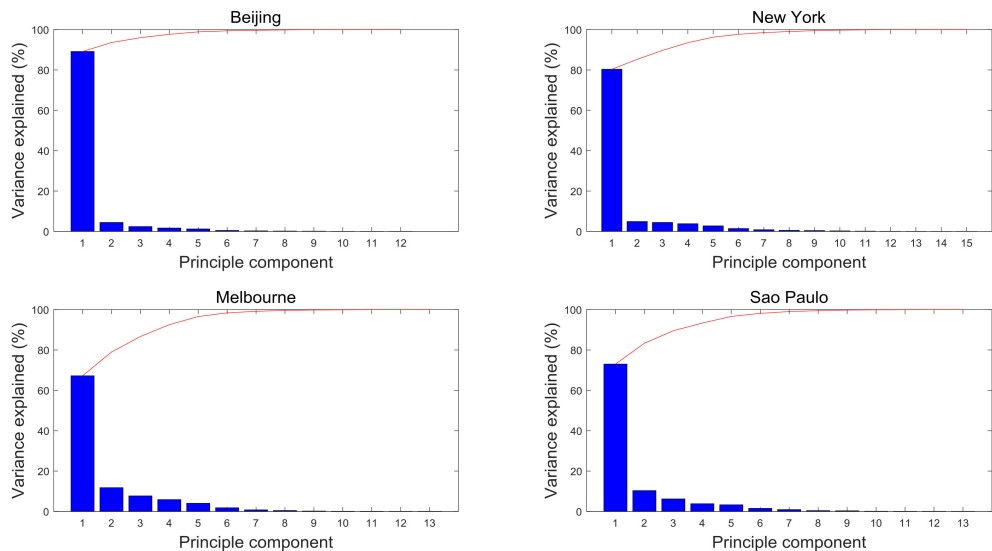

**Figure 7.** Scree plot of Beijing, New York, Melbourne, and São Paulo in principal component analysis (PCA) analysis.

**Table 4.** Component Matrix of Beijing, New York, Melbourne, and São Paulo

| Beijing | | New York | | Melbourne | | | | São Paulo | | |
|---|---|---|---|---|---|---|---|---|---|---|
| Component | PC1 | Component | PC1 | Component | PC1 | PC2 | PC3 | Component | PC1 | PC2 |
| SH2M | 0.899 | SH2M | 0.87 | SH2M | 0.847 | −0.298 | 0.198 | SH2M | 0.902 | −0.162 |
| EST | 0.973 | EST | 0.861 | EST | 0.81 | −0.056 | 0.458 | SP | 0.906 | −0.005 |
| T2M_DEW | 0.954 | T2M_DEW | 0.854 | T2M_DEW | 0.782 | 0.22 | 0.513 | EST | 0.881 | 0.21 |
| T2M_MAX | 0.932 | T2M_MAX | 0.856 | T2M_MAX | 0.774 | 0.415 | 0.321 | T2M_DEW | 0.857 | 0.38 |
| T2M | 0.975 | T2M | 0.859 | T2M | 0.765 | 0.518 | 0.004 | T2M | 0.84 | 0.459 |
| T2M_MIN | 0.985 | T2M_MIN | 0.861 | T2M_MIN | 0.741 | 0.517 | −0.255 | T2M_MIN | 0.816 | 0.455 |
| Lag 1 $x_{t-1}$ | 0.966 | Lag 1 $x_{t-1}$ | 0.862 | Lag 1 $x_{t-1}$ | 0.704 | 0.4 | −0.324 | Lag 1 $x_{t-1}$ | 0.78 | 0.383 |
| Lag 2 $x_{t-2}$ | 0.96 | Lag 2 $x_{t-2}$ | 0.854 | Lag 2 $x_{t-2}$ | 0.779 | −0.451 | 0.034 | Lag 2 $x_{t-2}$ | 0.928 | −0.152 |
| Lag 3 $x_{t-2}$ | 0.954 | Lag 3 $x_{t-2}$ | 0.834 | Lag 3 $x_{t-2}$ | 0.927 | −0.115 | −0.22 | Lag 3 $x_{t-2}$ | −0.713 | 0.214 |
| Lag 4 $x_{t-4}$ | 0.95 | Lag 4 $x_{t-4}$ | 0.93 | Lag 4 $x_{t-4}$ | 0.776 | −0.459 | 0.021 | Lag 4 $x_{t-4}$ | 0.813 | −0.417 |
| Lag 5 $x_{t-5}$ | 0.941 | Lag 5 $x_{t-5}$ | 0.959 | Lag 5 $x_{t-5}$ | 0.869 | −0.077 | −0.291 | Lag 5 $x_{t-5}$ | 0.923 | −0.153 |
| | | Lag 6 $x_{t-6}$ | 0.938 | Lag 6 $x_{t-6}$ | 0.924 | −0.259 | −0.15 | Lag 6 $x_{t-6}$ | 0.922 | −0.311 |
| | | Lag 7 $x_{t-7}$ | 0.961 | Lag 7 $x_{t-7}$ | 0.92 | −0.148 | −0.241 | Lag 7 $x_{t-7}$ | 0.795 | −0.448 |
| | | Lag8 $x_{t-8}$ | 0.962 | | | | | | | |
| | | Lag 9 $x_{t-9}$ | 0.965 | | | | | | | |

## 3.3. Parameters Setting and Forecasting Evaluation Criteria

The PCA-WT-BA-ELM is utilized for the solar radiation prediction in this paper. With the intention of verifying the superiority of the model, the study makes the comparison of solar radiation predictions and varieties of model settings. The comparison shown in Figure 8 contains four parts. In the first part, in order to test the performance of the prediction method, the single extreme learning machine (ELM), the least squares support vector machine (LSSVM), and the backward propagation neural network (BPNN) are used for comparison. In the second part, the single ELM, PSO-ELM, and BA-ELM are collected to prove the effectiveness of the optimization approach and further certify the superiority of BA-ELM. In the third part, BA-ELM and WT-BA-ELM are used to display the progress of the decomposition method WT. In the fourth part, WT-BA-ELM and PCA-WT-BA-ELM are compared in order to prove the necessity and capability of the dimensionality reduction method.

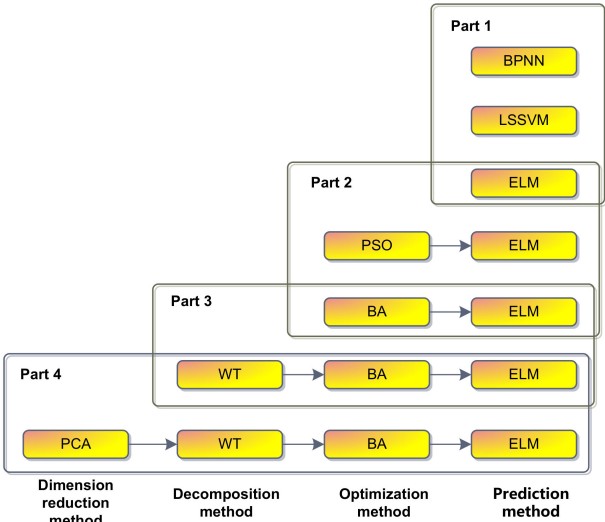

**Figure 8.** Framework of the solar radiation forecasting model comparisons.

Since the parameter settings may readily affect the prediction accuracy, it is prerequisite to define the comparison model's parameters, and the specifications are as displayed in Table 5.

**Table 5.** Parameters of three types of artificial neural networks

| Model | Parameters |
|---|---|
| BPNN | L = 10; learning rate = 0.0004 |
| LSSVM | L = 10; $\gamma$ = 50; $\sigma^2$ = 2 |
| ELM | L = 10; g(x) = 'sig'; |
| PSO-ELM | N = 10; N_iter = 500; c1 = c2 = 2; w = 1.5; rand = 0.8 |
| BA-ELM | N = 10; N_iter = 500; A = 1.5; $\gamma$ = $\theta$ = 0.9; R = 0.0001; F = [0, 2] |

L represents the hidden layer neuron number, $\gamma$ represents regularization parameter, σ2 represents kernel parameter, g (x) represents the hidden layer activation function, N represents Initial population size, N_iter is the maximum number of iterations, and c1 and c2 are acceleration factors. W is inertia weight, and rand is generated uniformly in the interval [0, 1]. A represents Initial Impulse volume, $\gamma$ and $\theta$ are the attenuation coefficient of the volume and the enhancement coefficient of the search frequency. R is impulse emission rate, and F represents the range of frequency. The values of each parameter in Table 5 are repeatedly adjusted through the simulation process to finally obtain a satisfactory value.

For the purpose of measuring prediction performance with effect, four commonly used error criteria were proposed to test the accuracy of all relevant models—mean absolute error (MAE), mean absolute percentage error (MAPE), the root mean squared error (RMSE), and the coefficient of determination $R^2$. The formulas are represented as follows.

$$
\begin{aligned}
\text{MAE} &= \tfrac{1}{n}\left|y_i - y_i^*\right| \\
\text{MAPE} &= \tfrac{1}{n}\sum_{i=1}^{n}\left|\frac{y_i - y_i^*}{y_i}\right| \times 100\% \\
\text{RMSE} &= \sqrt{\tfrac{1}{n}\sum_{i=1}^{n}\left|\frac{y_i - y_i^*}{y_i}\right|^2} \\
R^2 &= \frac{\left(n\sum_{i=1}^{n} y_i \times y_i^* - \sum_{i=1}^{n} y_i \sum_{i=1}^{n} y_i^*\right)^2}{\left(n\sum_{i=1}^{n} y_i^{*2} - \left(\sum_{i=1}^{n} y_i^*\right)^2\right)\left(n\sum_{i=1}^{n} y_i^2 - \left(\sum_{i=1}^{n} y_i\right)^2\right)}
\end{aligned}
\tag{9}
$$

where *n* is the number of training samples, and $y_i$ and $y_i$ * are actual and predicted values, respectively.

### 3.4. Solar Radiation Forecasting

#### 3.4.1. The Case of Beijing

The proposed model BA-ELM and its comparison models PSO-ELM, LSSVM, and BPNN are all implemented in MATLAB R2017a on a Windows 10 system. The results of Beijing solar radiation after prediction are presented in Figure 9. The error analysis and actual values of MAE, MAPE, RMSE, and $R^2$ are given respectively in Figure 10. Referring to Figure 10, Table 6, the following can be obtained:

(a) The MAE, MAPE, and RMSE of PCA-WT-BA-ELM are the minimum and the $R^2$ is the maximum, which demonstrates its performance sufficiently;

(b) The predicted carbon price curve is closest to the actual carbon price curve, which is better than the Single-LSSVM and Single-BPNN. Single ELM's MAE, MAPE, RMSE and $R^2$ surpasses Single-LSSVM and Single-BPNN, showing that Single-ELM has the best predictive performance. In addition, as can be discovered in Table 6, the learning speed of Single-ELM is the shortest, reflecting that in the part of prediction accuracy and learning speed, Single-ELM exceeds Single-LSSVM and Single-BPNN;

(c) When Comparing with single ELM, hybrid models (including PSO-ELM and BA-ELM) have smaller MAE, MAPE, RMSE, and larger $R^2$, which shows that it makes sense to optimize the ELM parameters. BA-ELM's MAE, MAPE, and RMSE are smaller, and BA-ELM's $R^2$ is larger than PSO-ELM's $R^2$, reflecting that BA-ELM is more precious in the whole, and BA is superior to PSO in the part of optimizing the parameter of ELM;

(d) After the comparison with BA-ELM, the predicted solar radiation curve of WT-BA-ELM is closest to the actual one. For that the solar radiation series is highly uncertain, nonlinear, dynamic, and complex, it may not be appropriate to predict straight without decomposition. It can therefore be seen that the MAE, MAPE, RMSE, and $R^2$ of WT-BA-ELM are better than BA-ELM;

(e) The predicted solar radiation curve of PCA-WT-BA-ELM is closer to the actual solar radiation curve than that of WT-BA-ELM. The MAE, MAPE, RMSE, and $R^2$ of PCA-WT-BA-ELM are better than WT-BA-ELM. All of this can verify the need to use PCA to reduce the dimensions of the BA-ELM input.

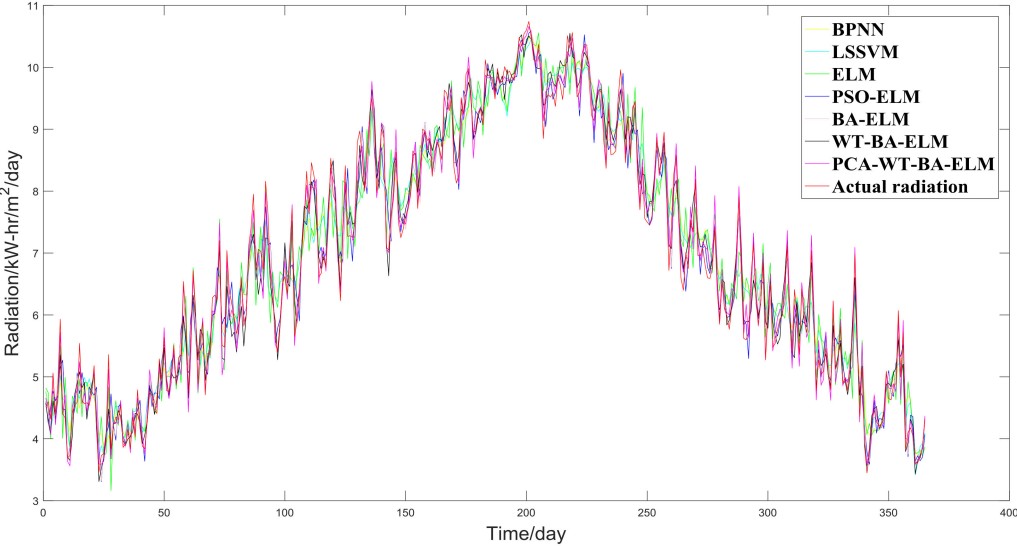

**Figure 9.** The forecasting results of Beijing solar radiation.

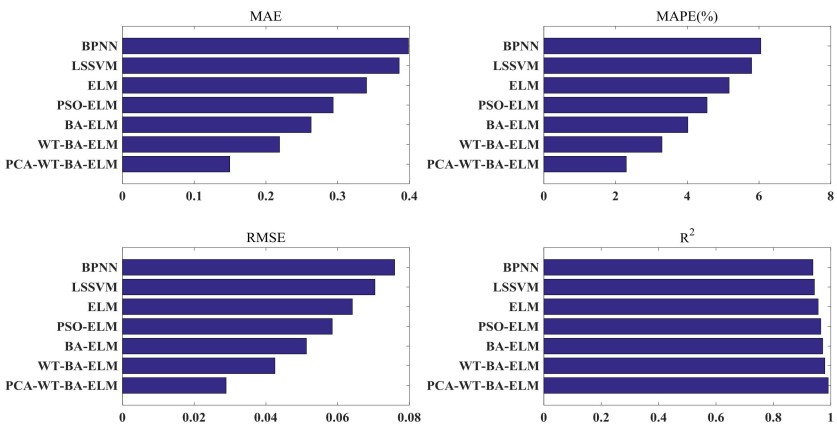

**Figure 10.** Error analysis of Beijing solar radiation forecasting.

**Table 6.** Learning speed comparation analysis of BPNN, LSSVM, ELM.

|  | **Training Time (s)** | **Test Time (s)** |
|---|---|---|
| BPNN | 2621.112 | 0.265 |
| LSSVM | 1784.593 | 0.153 |
| ELM | 19.341 | 0.001 |

### 3.4.2. The Case of New York, Melbourne, and São Paulo

In order to present the predicting capacity of the proposed model, the solar radiation of New York, Melbourne, and São Paulo are utilized. Figures 11–13 are the forecasting results, and Figures 14–16 are the error analysis of forecasting results. We are able to draw the conclusion that the hybrid model PCA-WT-BA-ELM has the most outstanding predictive ability, which is measured by the MAE, MAPE, RMSE, and $R^2$. It can forecast the solar radiation of different parts of the world.

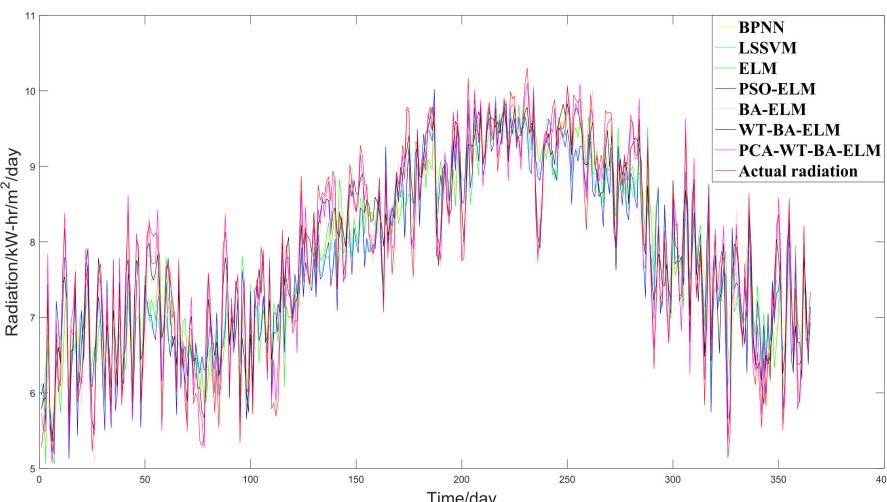

**Figure 11.** The forecasting results of New York solar radiation.

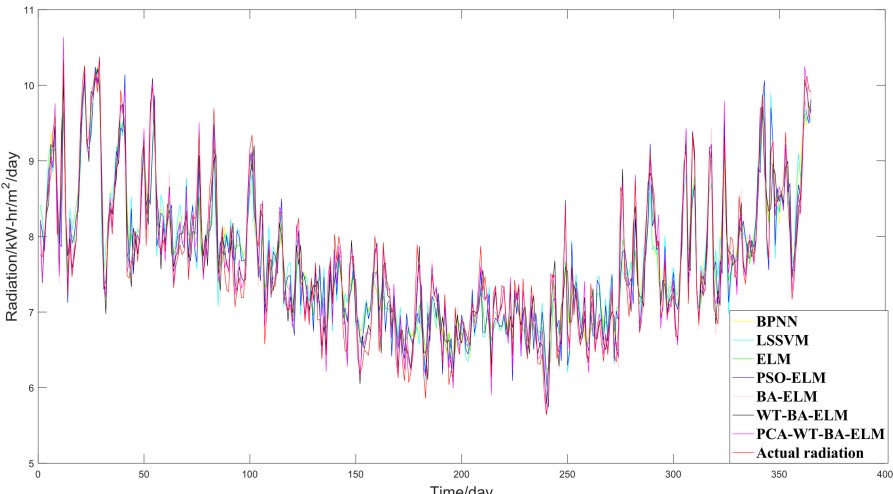

**Figure 12.** The forecasting results of Melbourne solar radiation.

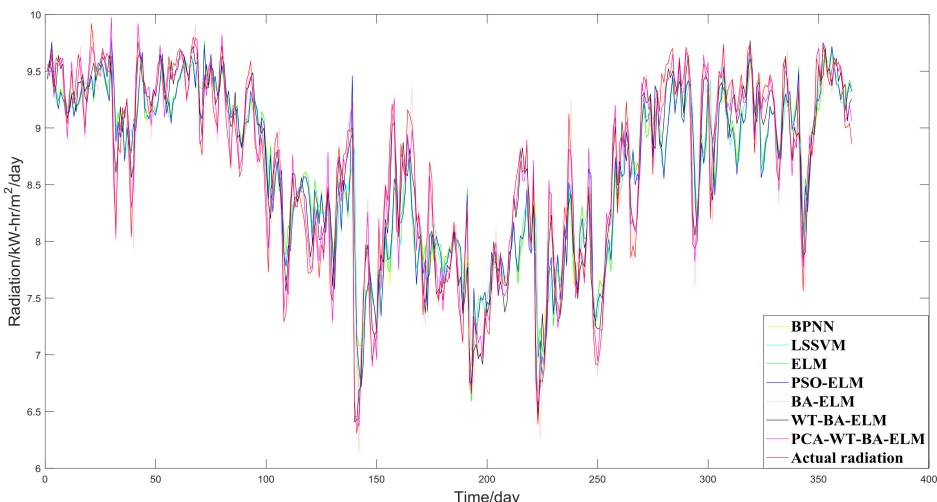

**Figure 13.** The forecasting results of São Paulo solar radiation.

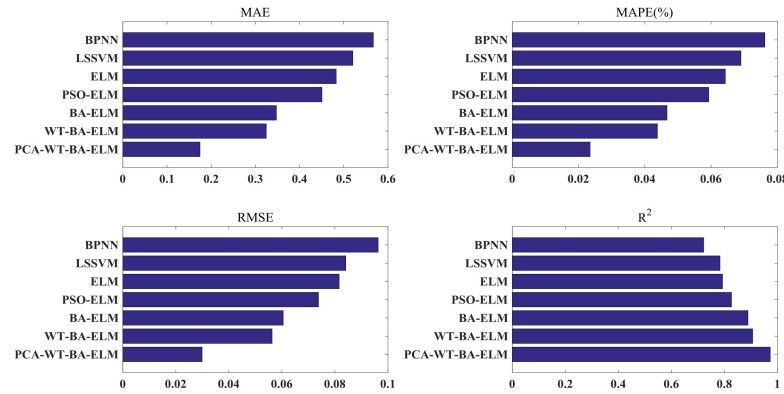

**Figure 14.** Error analysis of New York solar radiation forecasting.

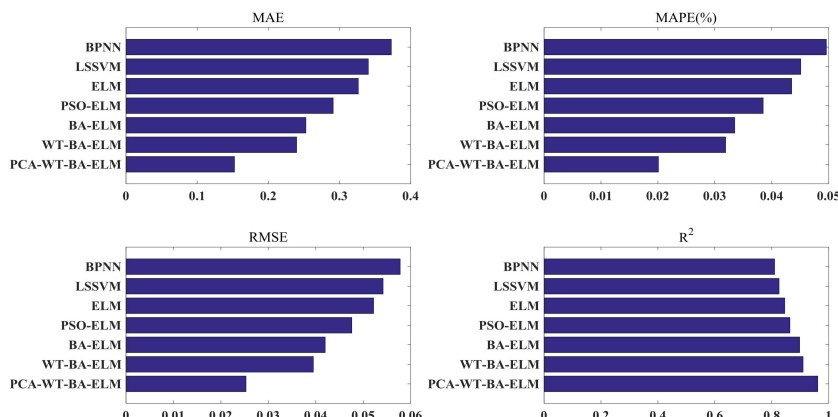

**Figure 15.** Error analysis of Melbourne solar radiation forecasting.

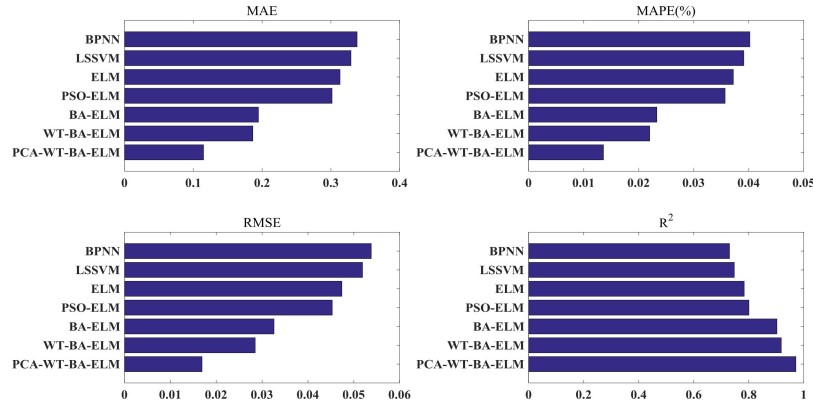

**Figure 16.** Error analysis of São Paulo solar radiation forecasting.

## 4. Conclusions

In this paper, a new hybrid model of extreme learning machine based on WT and PCA-based BA optimization algorithm is proposed for solar radiation prediction. The solar radiation series is divided into two parts: an approximate series (de-noised signal) and a detail series (noise). BA is employed to get the input weight matrix and the hidden layer bias matrix, which correspond to the ELM of the minimum training error. PACF is used to choose the lags of the approximation series, which are the inputs of BA-ELM. Diverse model and solar radiation series are employed in order to check the capability and effectiveness of PCA-WT-BA-ELM. In summary, according to the carbon price prediction results of Beijing, New York, Melbourne, and São Paulo, the following conclusions can be drawn:

(a) ELM is superior to BPNN and LSSVM in predicting accuracy and learning speed. Because the ELM parameter, which the users have to make appropriate adjustments of, is the just number of hidden nodes. After stochastically installing the input weight and the hidden layer deviation, the output weight of the ELM can be analytically determined by solving the linear system according to the Moore-Penrose (MP) generalized inverse idea.

(b) In terms of prediction precision, both BA-ELM and PSO-ELM are superior to ELM, and BA-ELM is better than PSO-ELM. Therefore, it makes sense to optimize the parameters of the ELM through optimization method, and BA is more competitive than PSO;

(c) The model using decomposition method WT-BA-ELM performs better than that without it, which means that the decomposition method is able to ameliorate the forecasting performance, and it is essential to denoise the solar radiation sequence through WT as its uncertain, nonlinear, dynamic and complex features;

(d)　Compared with the model not using dimensionality reduction method WT-BA-ELM, the model with PCA-WT-BA-ELM is superior. It shows that the dimension reduction method is able to enhance the forecasting performance, and it is a necessity to decrease the dimension of many input indicators of BA-ELM through PCA;

(e)　The PCA-WT-BA-ELM model is superior to other methods in solar radiation prediction in Beijing, New York, Melbourne, and São Paulo. It can therefore be inferred that the proposed hybrid model can be utilized to predict solar radiation in different parts of the world at the same time, and it greatly expands the application of the model.

This paper primarily studies solar radiation predictions that take consideration of meteorological indicators and historical solar radiation sequences, as well as many other factors affecting solar radiation, such as PM 2.5, PM10, $O_3$, and Air Quality Index (AQI). For the fact of the serious air pollution, these indicators are very significant for Beijing solar radiation prediction. Hence, there are still several directions to study in the future.

**Author Contributions:** X.Z. is responsible for the conceptualization, data collation, formal analysis, methodology and original draft writing of this article. Z.W. is responsible for the preparation, writing and editing of this article.

**Funding:** This work is supported by the Fundamental Research Funds for the Central Universities (Project No. 2018MS144).

**Acknowledgments:** Thank the North China Electric Power University Energy Economic Development Strategy Research Base for its support in the research of this paper.

**Conflicts of Interest:** The authors declare no conflict of interest.

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
