# Peer review of "A Hybrid Model Based on Principal Component Analysis, Wavelet Transform, and Extreme Learning Machine Optimized by Bat Algorithm for Daily Solar Radiation Forecasting"

_sustainability, doi:10.3390/su11154138_

Round 1
Reviewer 1 Report
@page { size: 210.01mm 297mm; margin: 20mm } p { margin-bottom: 2.47mm; line-height: 120%; background: transparent } a:visited { color: #800000; so-language: zxx; text-decoration: underline } a:link { color: #000080; so-language: zxx; text-decoration: underline }
Review report for manuscript sustainability-526379 entitled:
Title: A Hybrid BA-ELM Model Based on PCA and WT for Daily Solar Radiation Forecasting
n the manuscript sustainability-526379 entitled “A Hybrid BA-ELM Model Based on PCA and WT for Daily Solar Radiation Forecasting”, authors present a new hybrid model for solar irradiation forecasting based on extreme learning machine (ELM) optimized by the
bat optimization algorithm (BA) based on wavelet transform (WT) and principal component. They provide a detailed state of the art related to solar irradiation prediction models. They also report prediction for 4 cities over the world (different longitude and altitude). They, also, compared results to other models proposed previously in the literature.
Authors proposed a promising model for solar irradiation forecasting based on historical solar irradiation data and meteorological data. The novelty of work is moderate and the experimental details and results need to be clarified in more detail. In my opinion, after addressing the following comments carefully, the manuscript can be considered for publication in “The European Physical Journal Plus”.
Main shortcomings are as follow:
• The content is not expressed clearly and in prop English. English language and style should be improved because there are a lot o confusing and unclear sentences.
• Chapters “Methodology” and “The proposed model” can be combined in a single chapter without extensive descriptions of all well-known concepts.
• Only data for large cities are used for model testing. Influence of air pollution is not clear. Not included in the model. It will be interesting to include in the model and test location with different level of pollutants in the air
• In general, image descriptions in the manuscript are not clear. For example for Figure 1 missing explanation for figure labels
• not all labels in equations are explained and defined
• not all abbreviations are defined when first used. For example, DSR used but not defined in the first sentence of Introduction, AWOA, …
• Abbreviations are not consistent: PACF and PCAF
Abbreviation in the manuscript title is not good practice
there are several sentences that are repeated twice. For example at the first page: “Many researchers have given DSR predictions.”. The manuscript should be checked for that.
For some algorithms used, references not given.
In Abstract: “… distinct solar radiation series obtained by NARA ...”. I believe that the author meant “by NASA?
There are confusing sentence and terms. For example “approximate” and “detailed time series”. I believe that is more clear to say that for irradiation data noise is filtered.
In the Introduction the first paragraph should be supported by references.
In the Methodology:
only for PCA is specified software used for calculation. Should be given for all other calculations
Is there any explanation why for Bet Algorithm is used value 0.9 for γ and θ
the author stated: “ … BA has been utilized in various areas ...” Should be stated which one and supported by references.
reference for Moore-Penrose missing.
equations in ELM paragraph are not clear (without variables descriptions)
Empirical analysis:
Reference for source of irradiation and meteorological data should be given (web url or article ref.)
Is there any influence of location elevation on model prediction results?
Figure 4: It’s difficult to resolve data for all cities. It will be better to use subplots for different cities (similar to fig 5a). Also for x scale, it will be better to use years and months as labels.
Figure 7: Meaning of “r” line in the graph should be explained in Figure caption or text
It’s not clear why SP parameter is included in the calculation for Beijing if Pearson coef. for SP is -0.779 (less than limit 0.8)?
“… When compared with the original carbon price ...”. I believe it’s should be irradiation data. If there is a relation to a carbon price, it should be explained in more details.
Table 4: the not clear meaning of “-” mark at the end of several rows
Figure 8: Instead of EA should be BA?
It’s not clear how the parameters listed in Table 5 are chosen, are they optimized somehow?
Figure 10 and Table 6 presents the same results. One can be omitted.
Results for all cities should be compared to each other. What is the difference?
Reviewer 2 Report
The paper addresses the development of a new method for prediction of solar radiation which is valuable for PV systems. However, it does not address the biggest issue: day-ahead and hour-ahead forecasting. As a result the paper is unacceptable in the current form. I suggest the following changes:
Use the data from 2014 - 2016 to create the model .
Apply the model to existing 2017 data to see how accurate are the predictions. This will give an indication of the efficiency of the method for day-ahead and hour-ahead forecasting.
Other Changes:
Please have the article read by a native English speaker. There are many instances of wrong usage of words and grammatical errors.
Line 28: Solar conversion is not a foremost parameter in meteorology. Please rephrase the sentence.
Line 29 No upper case for "Daily". Similar error in many places.
Line 35: DSR expand the first time
Line 43-44: Sentence repetition
Line 48: Space before 2015
Line 108: Solar radiation is not a chaotic feature. It has a well-defined pattern which is influenced by many parameters.
Line 282 - 288: Details missing on the source of the data and its limitation. This is a very important information which needs to be provided.
Round 2
Reviewer 1 Report
The authors have taken the reviewers' comments seriously and revised the manuscript accordingly. However, there are still several spelling and typing errors that, I believe, can be picked up in typesetting. Because of bad poor formatting, the content of Table 4 is still confusing (values/digits separated in two rows, minus sign separated from value.)
In my opinion, the revised manuscript could be considered for publication in the Sustainability journal.
Author Response
The paper has been revised, please check.

Reviewer 2 Report
Needs a language check before accepting.
Author Response
The paper has been revised, please review it.
